# Improving the acceptability of social robots: Make them look different from humans

**Tatjana A. Nazir** *, **Benjamin Lebrun**, **Bing Li**

University Lille, CNRS, UMR 9193 –SCALab—Sciences Cognitives et Sciences Affectives, Lille, France

* tatjana.nazir@univ-lille.fr

## Abstract

The social robots market will grow considerably in the coming years. What the arrival of these new kind of social agents means for society, however, is largely unknown. Existing cases of robot abuse point to risks of introducing such a̲rtificial s̲ocial a̲gents (ASAs) without considerations about consequences (risks for the robots and the human witnesses to the abuse). We believe that humans react aggressively towards ASAs when they are enticed into establishing dominance hierarchies. This happens when there is a basis for skill comparison. We therefore presented pairs of robots on which we varied similarity and the degree of stimulatability of their mechanisms/functions with the human body (walking, jumping = simulatable; rolling, floating = non-simulatable). We asked which robot (i) resembled more a human, (ii) possessed more "essentialized human qualities" (e.g. creativity). To estimate social acceptability, participants had also (iii) to predict the outcome of a situation where a robot approached a group of humans. For robots with simulatable functions, rating of essentialized human qualities decreased as human resemblance decreased (jumper < walker). For robots with non-simulable functions, the reversed relation was seen: robots that least resembled humans (floater) scored highest in qualities. Critically, robot's acceptability followed ratings of essentialized human qualities. Humans respond socially to certain morphological (physical aspects) and behavioral cues. Therefore, unless ASAs perfectly mimic humans, it is safer to provide them with mechanisms/functions that cannot be simulated with the human body.

## Introduction

Social companion robots are physically embodied artificial intelligence (AI) systems designed to interact and communicate with humans. Currently, these AI-based machines can carry out tasks such as guarding the house, helping in the kitchen, entertaining, teaching, or assisting individuals with needs or disabilities [e.g. 1–5]. A special class of social companion robots are also devised to act as surrogates for human sexual partners. Over the past few years, the number and the variety of social companion robots has grown drastically. Market research companies forecast that between 2021 and 2026 the social robot market will grow by a compound annual growth rate (CAGR) of more than 14%. What the arrival of these new kind of social agents will mean for society, however, is largely unknow.

**Data Availability Statement:** Data are available at: https://osf.io/pkhj4/.

**Funding:** This project has received funding from the MEL (Métropole européenne de Lille) and the I-

SITE ULNE (Université Lille Nord-Europe). Grant awarded to TAN (n°: R-Talent-20-006-Nazir).

**Competing interests:** Authors declare that they have no competing interests.

From an ethological point of view, social behavior serves a function and has specifically evolved to contribute to survival of species [6,7]. Individuals cooperate, for instance, to facilitate the acquisition of food and to avoid predators. On this background, the design of social robots should take into account that certain morphological (physical aspects) and behavioral cues may trigger primary social responses in humans. The "*baby schema*" [8], for instance, is one of the best known of these biological relevant cues. Juvenile animals including humans have higher foreheads, larger eyes, and smaller mouths than their adult counterparts. These features (which are widely exploited in product design, e.g. [9]), give them their characteristic "cuteness" that triggers nurturing behavior in adults [10]. Likewise, the "*attachment-behavior*" system that is believed to have evolved for protection [11] and social/cultural learning [12] is sensitive to certain cues and can be activated in a given context [e.g. 13]. Among such cues/contexts are proximity and the sharing of a common goal [11,14]. Since attachment is not limited to members of the same species (see human-animal relationships; [12]), it is likely to also develop in human-social AI relationships. This, in turn, could be leveraged by manufacturers, e.g. via software support, upgrade costs, and in-app purchases (see [15]). Our tendency to *anthropomorphize*, i.e., to attribute animacy and social meaning to non-human agents (see e.g. [16]) is another critical feature of human social behavior. Humans look at others in the light of their own mental properties in order to predict and understand them (c.f. the "intentional stance"; [17]). According to [18], this tendency to infer thoughts, feelings and intentions from behavior is a default mechanism, selected by biological and cultural evolution. The anthropomorphization of robots, which is facilitated by human-like traits (e.g. [19,20]), could raise expectations that the artificial social agent (ASA) might not be able to meet (see [21]). This creates frustration that could lead to aggressive reactions [22]. A further critical aspect of human behavior is our evolved mechanism of *disease-avoidance*, which can lead to prejudice against individuals with physical disabilities (see [23,24]). Since infectious diseases are associated with particular physical characteristics (e.g. discoloration of body parts, lesions, behavioral anomalies, etc.), Park and colleagues speculate that humans have evolved mechanisms that elicit certain emotions/behavior (e.g. disgust, fear, avoidance) when these traits are perceived. The accidental presence of such traits in an ASA's design could thus trigger undesirable human behavior. Finally, the tendency of social animals (including humans) to organize into *dominance hierarchies* is also a factor worth scrutinizing. A dominance hierarchy (c.f. "pecking order"; [25]) is a ranking system that is established by members of animal social groups. Physically "formidable" (i.e. larger; stronger; more powerful) individuals often dominate weaker ones [26,27]. These higher-ranking individuals typically display an assertive attitude that can be accompanied by intimidation behavior and aggression [28]. In humans, the social rank is additionally determined by prestige, i.e. status that comes from excelling in valued domains (c.f. cultural know-how and guidance) [29]. The human tendency to establish social hierarchies is likely to also be at work when interacting with ASAs. While cultural know-how (e.g. excelling in the Chinese board game "Go") could be a decisive factor in assigning a social rank to robots (c.f. prestige), morphological aspects (c.f. formidability-cues) could be potentially important as well. One fundamental societal question for the design of ASAs is therefore that of the social rank that we want to assign to companion robots—knowing that this rank may partly determine our attitude towards them.

From the manipulability of human social behavior by biologically relevant cues to more complex factors related to attachment and the establishment of dominance hierarchies, there are thus a number of significant issues that could arise from the interactions between humans and ASAs. For example, there are now a series of reports about verbal and physical abuse of AI and social robots by humans, including children (for a recent summary see e.g. [30]). Along this line, Amazon and Google have adapted their AI-Digital assistants (Alexa and Google

Assistant) to allow countering the poor verbal manners of children and adolescents when addressing the devices. Obviously, artificial social agents are not sentient and therefore do not suffer from such negative social behavior. The problem lies elsewhere, however. Bullying behavior does not only hurt the victim but has also a negative impact on non-bullied witnesses. Studies on the effect of bullying at the workplace, for instance, have testified a higher risk of depression and an increase of subjective stress in individuals that witnessed bullying behavior towards others (e.g. [31–33]). Similarly, studies with adolescents show that observing bullying at school predicts mental health risks (e.g. [34]). Critically, as demonstrated by Bartneck and Keijsers [30], abusing a robot is considered as immoral as abusing a human. Therefore, with regard to the consequences of witnessing abuse, it does not really matter that the robot is non-sentient.

If interacting with ASAs can trigger reprehensible human behavior, there is reason to be vigilant. To ensure the development of morally and legally acceptable social environments, it is thus important to know how introducing ASAs into society will affect human conduct and behavior. The present study seeks to contribute to such an endeavor, by focusing on potential links between a robot's physical appearance and its acceptability.

As pointed out by Miklosi and Gacsi [7], in nature, evolution ensures a correspondence between function (e.g. vision) and mechanisms (e.g. eyes) with the former constraining the latter. Unnecessary mechanisms will be selected against. Hence, the blind mole rat that stays underground most of its life (without needing vision) has no functional eyes ([7], p. 3). For this reason, morphological features such as eyes, mouth, or hands are interpreted as signaling the presence of specific functions. Robots with lips/mouth, for example, can prompt humans to expect language skills; robots with five-finger hands can prompt expectations about manual dexterity, etc. (see e.g. [35]). Certain mechanism/functions can also trigger more complex processes, such as the previously described disease-avoidance behavior or the tendency to establish dominance hierarchies. Hence, an ASA with (only) three fingers on its hands (a potential trigger for disease-avoidance behavior), which thereby suggests poorer performance in motor tasks (a potential trigger for assigning a lower rank in the dominance hierarchy), may not find immediate social success. Our hypothesis is that resemblance to humans in terms of mechanisms/functions is a crucial modulator of human social behavior towards robots. In fact, when the physical appearance of an ASA resembles that of a human, humans can use embodied mental simulations (see e.g. [36]) to estimate and compare the robot's capabilities (functions) relative to themselves. If in doing so, they judge (whether rightly or wrongly) that the ASA is less good (e.g. because the *human* hand is less efficient with three than with five fingers), the robot will be considered as having a lower rank. This downranking could come with dominance-accompanying behavior (intimidation or aggression). However, if such a comparison is precluded because the ASA is equipped with mechanisms/functions that are not in the human repertoire (e.g. mechanisms that allow displacing objects by levitation) such a hierarchical downranking of the artificial agent will not automatically occur. Our prediction is therefore that unless they are perfect human copies, ASAs with functions that can be simulated with the human body will be more prone to abuse than those with functions that cannot be simulated.

To test this hypothesis, we created a series of cartoon drawings of ASAs (note that a recent meta-analysis on the effect of anthropomorphism in human-robot interaction [37] found no difference between the use of depictions of robots as opposed the use of real robots on performance) in which we varied (i) the degree to which their physical appearance allowed embodied mental simulations (simulatable vs. non-simulatable functions) and (ii) the degree to which the mental simulations signaled more or less efficient performance compared to humans (operationalized by ratings of the degree of human resemblance). In addition, (iii) for each robot we determined the level at which it is perceived as possessing qualities that are

considered as characteristic of human nature (c.f. [38]). Since the denial of human nature (c.f. dehumanization) is an enabler of instrumental violence between humans (see e.g. [39]; for a review see [40]), the estimation of the degree to which our robots are perceived as possessing human qualities can be taken as further predictor of the likelihood of abuse. We also asked a group of participants to predict among four options the outcome of an encounter between the robots and a group of people: the robot will either be included, ignored, excluded, or attacked by the group. The last two options were considered as indicator of abuse.

## The study

### Physical characteristics of the ASA's

Robot stimuli were assembled using 3 components (see Fig 1) created with Apple Inc. Keynote: (i) A (grey) torso with a schematic (red) face with two eyes and a mouth, and elements that represent (ii) the lower and (iii) the upper extremities. The trunk/face is identical for all agents. However, there are four types of lower and upper extremities. *Lower extremities*: (i) two legs on feet (function: walking). (ii) One single leg with a foot (function: jumping). (iii) A chassis with three wheels (function: rolling.) (iv) A semi oval element (function: floating). *Upper extremities*: (i) hands with five fingers (function: gripping). (ii) Clamp-type pliers (function: pinching). (iii) magnetic-like horse hooves (function: attracting). (iv) parabola-like end parts (function: hovering). Note, for both extremities elements described under (i) and (ii) depict functions that can be simulated with the human body, while (iii) and (iv) cannot. Walking on two legs is most similar to humans' locomotion mode. Jumping on one leg is feasible with the human body but is less efficient than walking on two legs. However, rolling on a three-wheeled chassis and floating are not feasible with the human body alone (note though that the rolling function is something humans are quite familiar with from the use of vehicles, while the

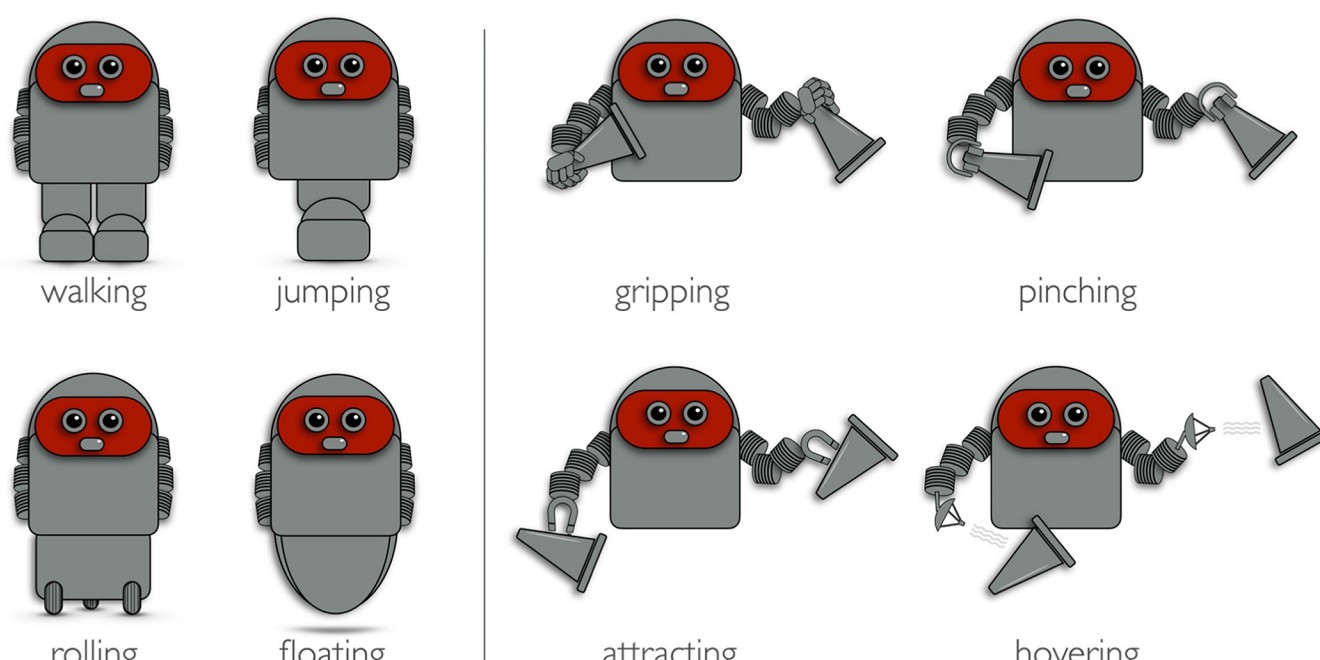

**Fig 1. Robot stimuli.** The four types of robots' lower and upper extremities and their functions. Left: Two legs on feet (walking); One leg with a foot (jumping); a chassis with three wheels (rolling); a semi oval element (floating). Right: Hands with five fingers (gripping); Clamp-type pliers (pinching); magnetic-like horse hooves (attracting); parabola-like end parts (hovering).

floating function is much less familiar). Similarly, gripping objects with five fingers is most similar to how humans manipulate objects. Pinching is also feasible with a five-fingered hand but is less efficient. By contrast, attracting objects like a magnet attracts special metals (familiar), and hovering objects around (less familiar) are not feasible with the human body alone.

Participants were informed though that when measured in terms of distance travelled per unit of time, all four locomotion modes resulted in the same speed. The efficiency of the four end parts of the arms for, e.g., collecting and arranging items was also the same. This was done to encourage participants not to base their ratings on assumptions about differences in robot performance.

The 2×4 robot extremities, allowed creating 16 different agents with distinct physical characteristics. The four robots that consistently combine functions that can be simulated with the human body (walking combined with either gripping or pinching; jumping combined with either gripping or pinching) and the four robots that consistently combine functions that cannot be simulated with the human body (rolling combined with either attracting or hovering; floating combined with either attracting or hovering) were the target robots for testing our hypothesis. The remaining eight robots that combine one function that can be simulated with the human body and one that cannot served as filler (interpretation of the simulatability of functions of these robots is not straightforward).

## Estimation of the human resemblance, of the possession of essentialized human qualities, and of the simulatability of functions

Short video clips served to familiarize participants with mechanisms/functions of the different robots (see method section). Using a two-by-two comparisons of all robots, we then asked groups of participants to determine which of the two robots (i) looked more like a human (human resemblance), and (ii) possessed more of a particular set of essentialized human qualities. The five qualities that we tested were drawn from the work on essentialist beliefs about human personality by Haslam, et al. [38], and are 'creativity', 'imaginability', 'independence', 'intelligence' and 'talkativeness'. Essentialist beliefs refer to the practice of regarding things (e.g. a trait) as innate and biological based rather than acquired (see e.g. [41]). According to [38] (see also [42,43]) personality traits are essentialized if they are considered as aspects of human nature. In their study, the five selected qualities were among the highest essentialized human personality descriptors out of a total of 80 tested items. Finally, to confirm our assertion about the simulatability of the 2×4 functions, a group of participants were also asked to indicate the degree of simulatability of the different functions with their own body on a 7-point Likert scale (1 = not simulatable; 7 = simulatable).

For the four robots with consistent simulatable functions, we expected that human resemblance scores would be lower for functions that are less efficient when performed with the human body (e.g. jumping/pinching < walking/gripping). For the four robots with consistent non-simulatable functions humans resemblance scores should generally be lower than for robots with consistent simulatable functions. Finally, if the attribution of essentialized human qualities depends on the degree of physical resemblance to humans, the two measures should correlate.

## Determining social acceptability

Using short video clips, participants were first familiarized with mechanisms/functions of the different robots (see material and method section). The acceptability of the ASAs was then estimated by exposing the participants to video clips depicting a social situation for which they had to predict the outcome. For this, a robot was shown approaching a group of people

consisting of either a group of children, teenagers, adults or elderly people. The clip ended before the robot reached the group and participants were asked to choose among 4 possibilities (presented as simple cartoons with verbal labels; see material and method section) the most likely outcome of the encounter. The options were: the robot will be i) 'included', ii) 'ignored', iii) 'excluded', or iv) 'attacked' by the group.

We expected that humans resemblance scores predict acceptability (operationalized by the 'include' option) only for robots with simulatable functions. Moreover, when robots with simulatable functions are not accepted, the 'exclude' and 'attack options, which we take as indicator of abuse, will be used more frequently than the ignore option compared to robots with non-simulatable functions.

## Material and method

### Ethics statement

All experiments were conducted in accordance with the ethical standards laid down in the 2013 Declaration of Helsinki and were approved by the local ethical committee (Comité d'éthique en sciences comportementales (ref.: 2021-467-S90). Testing was done online via an application hosted on Pavlovia servers (http://pavlovia.org) allowing data to be secured and anonymized [44]. Participants gave their informed consent by clicking on a pre-specified button on the screen with the information letter. Minors were tested following caregivers consent and the minor's independent decision to participate.

### Design, procedure, and participants

**Familiarization with the robot stimuli.** All participants were first familiarized with the robot-stimuli and the functions of the 2×4 different extremities. For this, each of the eight images presented in Fig 1 were shown individually together with a corresponding verbal description (e.g. "this robot has two feet"; "this robot attracts objects with magnetic hands"). Participants then saw the different functions of the extremities, using three-frame (for upper extremities) and four-frame clips (for lower extremities; see Fig 2). Participants were informed that even if the robots' extremities were different, all agents had the same level of performance (e.g., the same locomotion speed and the same capacity to collect or arrange items). This was done to prevent ratings from being based on beliefs about performance.

**Estimates/Ratings.** Estimations of ASA's (1) human resemblance and (2) essentialized human qualities were performed by two different groups of participants. A third group of participants determined (3) the social acceptability of the ASA's and rated (4) the simulatability of functions. Although simulatability of functions could be inferred from the physical appearance of the ASA's, the latter rating was done for the sake of completeness. The four procedures will be described below.

1. <u>Estimation of human resemblance:</u> Resemblance to human was defined as "what looks like a human". Using a two-by-two comparison, participants were requested to decide which of two robots resembled more a human. Robots were presented side-by-side on the computer screen and a decision was made by clicking with the computer mouse on one of the two agents. Participants' response triggered the ensuing trial. Each robot was shown in combination with each of the other robots (i.e., a total of 120 comparisons for the 16 different robots). For a given robot, the degree of human resemblance was calculated as: the number of times the robot was chosen divided by the total number (i.e. 15) of pairwise comparisons. It thus varies between 0 and 1. A rating session took less than 20 min. Twenty-five participants (11 females, 14 males; mean age = 27.28 years; range: 20–35 years) were recruited via

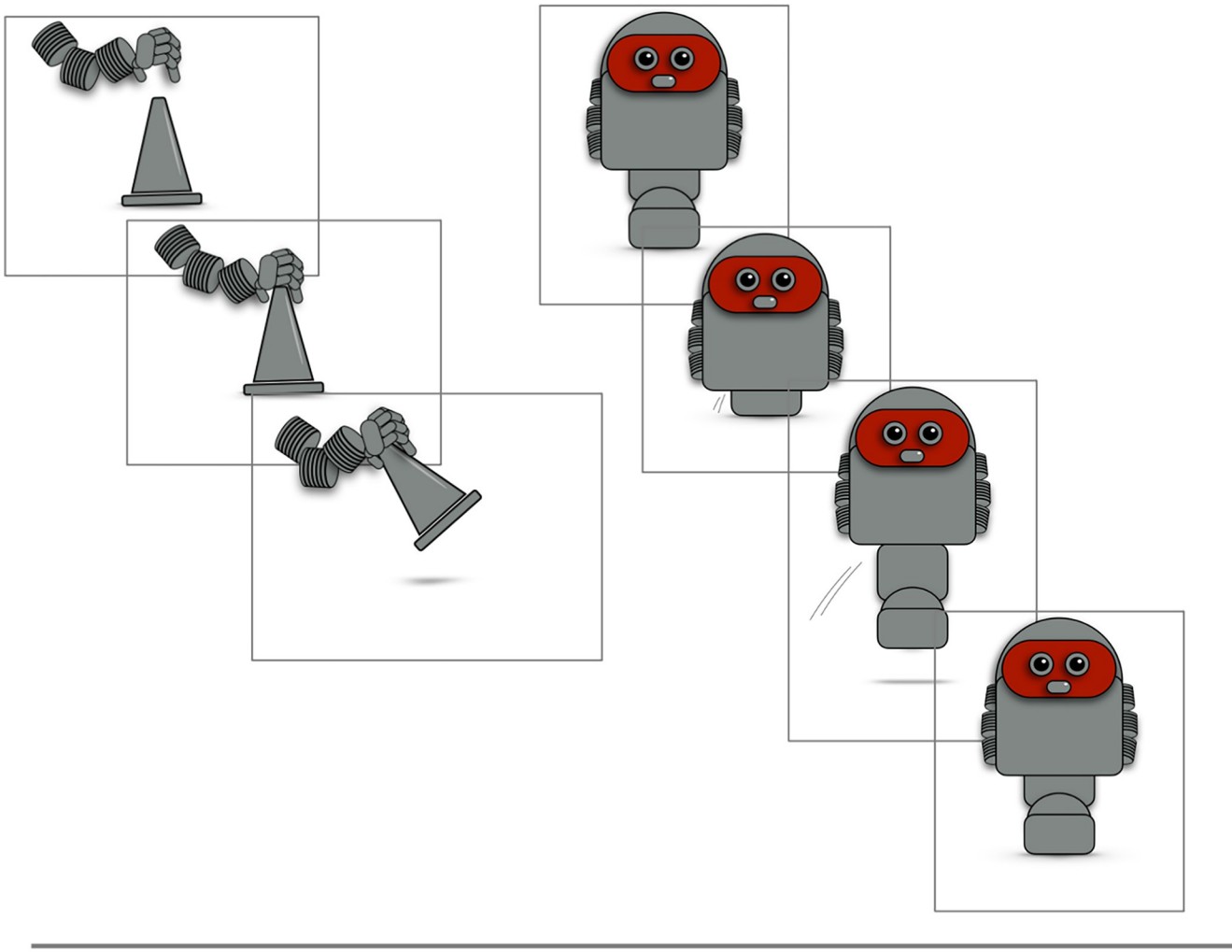

**Fig 2. Example of image frames for the clips.** Examples for the three- and four-image frames used to illustrate the functions of the upper (gripping) and lower (jumping) extremities of the robots as a function of time.

the Prolific website (https://www.prolific.co). They received a compensation of 7.50€ per hour. Data of one participant had to be discarded because of technical problems, leaving a total of 24 participants

2. <u>Estimation of the degree of essentialized human qualities</u>: Using the same two-by-two comparison, participants were requested to decide which of two robots expressed more of a given quality. Essentialized human qualities were 'independence', 'intelligence', 'imaginability', 'creativity', and 'talkativeness'. Recall that in the study by Haslam et al. [38] these five qualities were among the highest essentialized human personality descriptors out of a total of 80 tested items. Prior to the experiment, participants were given a definition of the qualities: Independent = "not to depend on another to live; to be autonomous"; Intelligent = "who has the ability to know and understand; who can think"; Imaginative = "who can imagine easily; who is inventive; who is carried away by his imagination"; Creative = "who has original ideas; who is inventive"; talkative = "who likes to talk; who talks a lot". Each

robot was shown in combination with each of the other robots (i.e., a total of 120 comparisons for the 16 different robots). For a given robot, the degree of perceived quality was calculated as: the number of times the robot was chosen divided by the total number (i.e. 15) of pairwise comparisons. It thus varies between 0 and 1. Ratings were preceded by 10 practice trials. A rating session took less than 20 min. A total of 48 participants (35 females, 13 males; mean age = 29.29 years; range: 10–81 years) rated the degree to which the robots possessed the different essentialized human qualities. Participants were recruited via social networks, they did not receive any compensation for completing the task. Note that ideally the same participant should have rated each of the five qualities. However, a pilot test revealed that the task was very repetitive and therefore there was a risk that it would not be performed correctly. We therefore asked participants to rate more than one quality only if they wished to do so. At the end, each quality was rated by a total of 25 participants.

3.  Measure of social acceptance. Prior to each trial, participants were first informed (by written text on the screen that a specific robot was going to approach a specific group of humans (e.g., "A robot on wheels with clamp-type pliers approaches a group of elderly people. What will these people do?"). Participants then viewed a short video clip in which a robot was seen approaching from the top left corner of the screen towards a group of four humans, displayed as black silhouettes (purchased from https://fr.123rf.com; see Fig 3 top panel) in the center of the screen. The group represented either children, teenagers, adults or elderly individuals. The robot was about half the size of an adult and 2/3 the size of a child. The clip showed the robot's mode of locomotion, but the upper limbs had been kept motion-less to prevent the robot from appearing agitated. The clip ended before the robot reached the group, and participants were asked to choose from four possibilities which they thought was the most likely outcome of the situation: (i) the robot is invited to join the

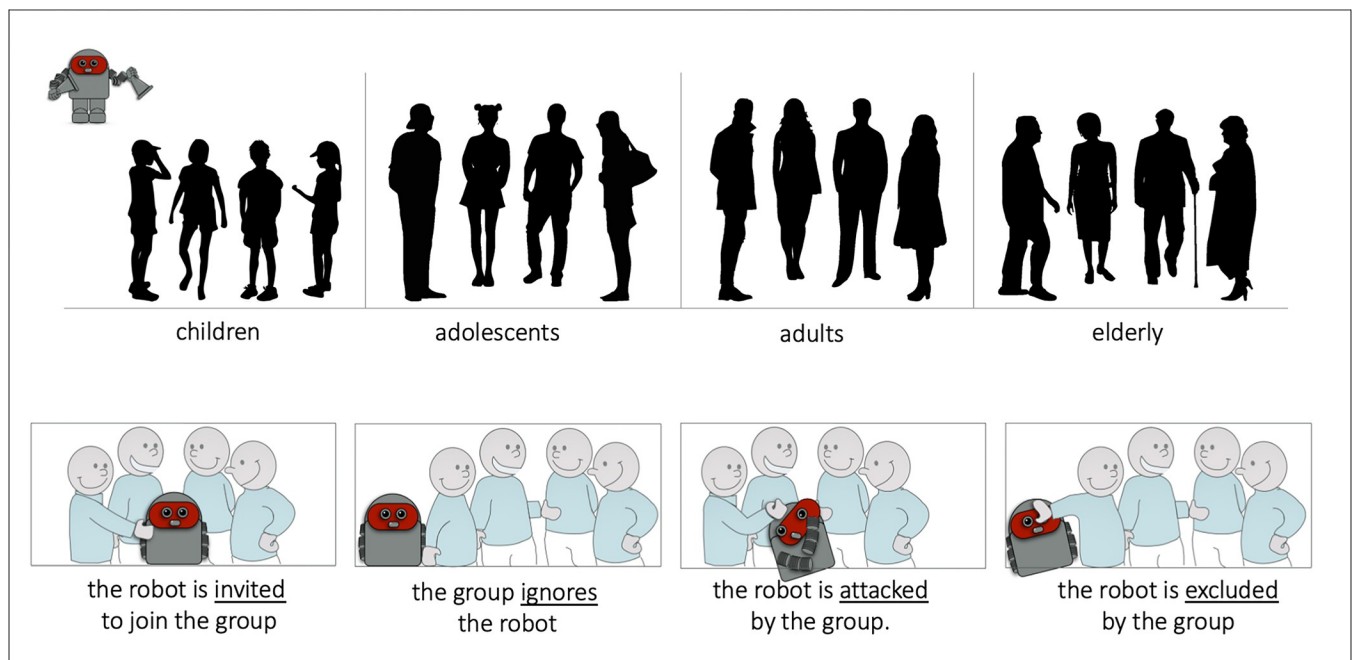

**Fig 3. Stimuli for the assessment of social acceptance.** (Top): The four groups of people (children, adolescents, adults, elderly) that the robot is approaching. On the left upper corner is an approaching robot. (Bottom): The four response options with the written descriptions (in the text the labels are referred to as (from left to right): "include"; "ignore"; "attack"; "exclude").

group, (ii) the group <u>ignore</u> the robot, (ii) the robot is <u>excluded</u> by the group or (iv) the robot is <u>attacked</u> by the group. Response options were given with simple line drawings of 4 ageless and genderless humans standing in a semi-circle smiling and chatting (Fig 3 bottom panel; modified from http://www.clipartsuggest.com/no-pigeonholes-please-whizwoman-s-blog-pyGXSg-clipart/). All four response options used the same drawings except for the arm/hand of one 'acting individual'. For the 'invite' option, the robot is shown standing in the center of the semi-circle and the acting person is placing its hand on the robot's shoulder. The 'ignore' option draws the robot outside of the semicircle, close to but slightly behind the group. The acting individual does not touch it. For the 'exclude' option, the acting individual is seen pushing the robot (hand in the robot's face) to prevent the robot from entering the semi-circle. The 'attack option shows the robot broken (scrambled elements) in the center of the semi-circle and the acting individual puts its fist on it. The four response options were displayed four seconds after the end of the video-clip to assure that participants focused their attention on the human-robot interaction before responding. The experiment was preceded by four training trials and took less than 30 min. Thirty-five participants (19 females, 11 males, 1 other; mean age = 24.7 years; range: 20–35 years) were recruited via the Prolific website (https://www.prolific.co). Data of four participants had to be discarded because of technical problems or because they did not finish the experiment, leaving a total of 31 participants. The same participants also rated the simulatability of the robots function (see below). For the two tasks, they received a compensation of 7.50€ per hour.

4. <u>Rating of simulatability</u>. At the end of the experiment on social acceptance, participants were asked to determine, on a 7-point Likert scale, to what extent they believe they are capable of performing the functions of each of the 2×4 extremities of the robots (1 = "I cannot at all" to 7 = "I am fully capable").

## Results

### 1) Simulatability and human resemblance

Simulatability and the degree of human resemblance were used to characterize the robots physical appearance. Simulatability was defined by the ways a robot was made, i.e. robots with consistent simulatable functions (SIM) combines upper/lower extremities with SIM functions ((jumping-pinching), (jumping-gripping), (walking-pinching), (walking-gripping)); robots with hybrid functions (h-SIM) combine SIM and non-SIM functions; robots with consistent non-simulatable functions (non-SIM) combine upper and lower extremities with non-SIM functions ((floating-hovering), (floating-attracting), (rolling-hovering), (rolling-attracting)). The three types of robots were treated as an ordinal scale (SIM, h-SIM, non-SIM). The degree of human resemblance is a continuous variable that varies between 0 and 1. Note that although our analyzes included all three types of robots (n = 16), we focus on comparing the 2×4 robots with consistent SIM and non-SIM functions in particular, as interpreting the results for hybrid robots is not straightforward.

**Simulatability of functions.** Screenings for the degree of stimulability of functions of the upper extremities confirmed that "hovering" and "attracting" were considered as non-simulatable (non-SIM) (median = 1 in both cases) while "pinching" and "gripping" as simulatable functions (SIM) (median = 7 in both cases). In the same way, medians for ratings for the lower extremities "hovering" and "rolling" (non-SIM) were 1 while the medians for "jumping" and "walking" (SIM) was 6 and 7.

**Degree of human resemblance.** Fig 4 shows boxplots of the proportion of human resemblance estimations for the 16 individual robots. Data are ordered from the lowest to the highest

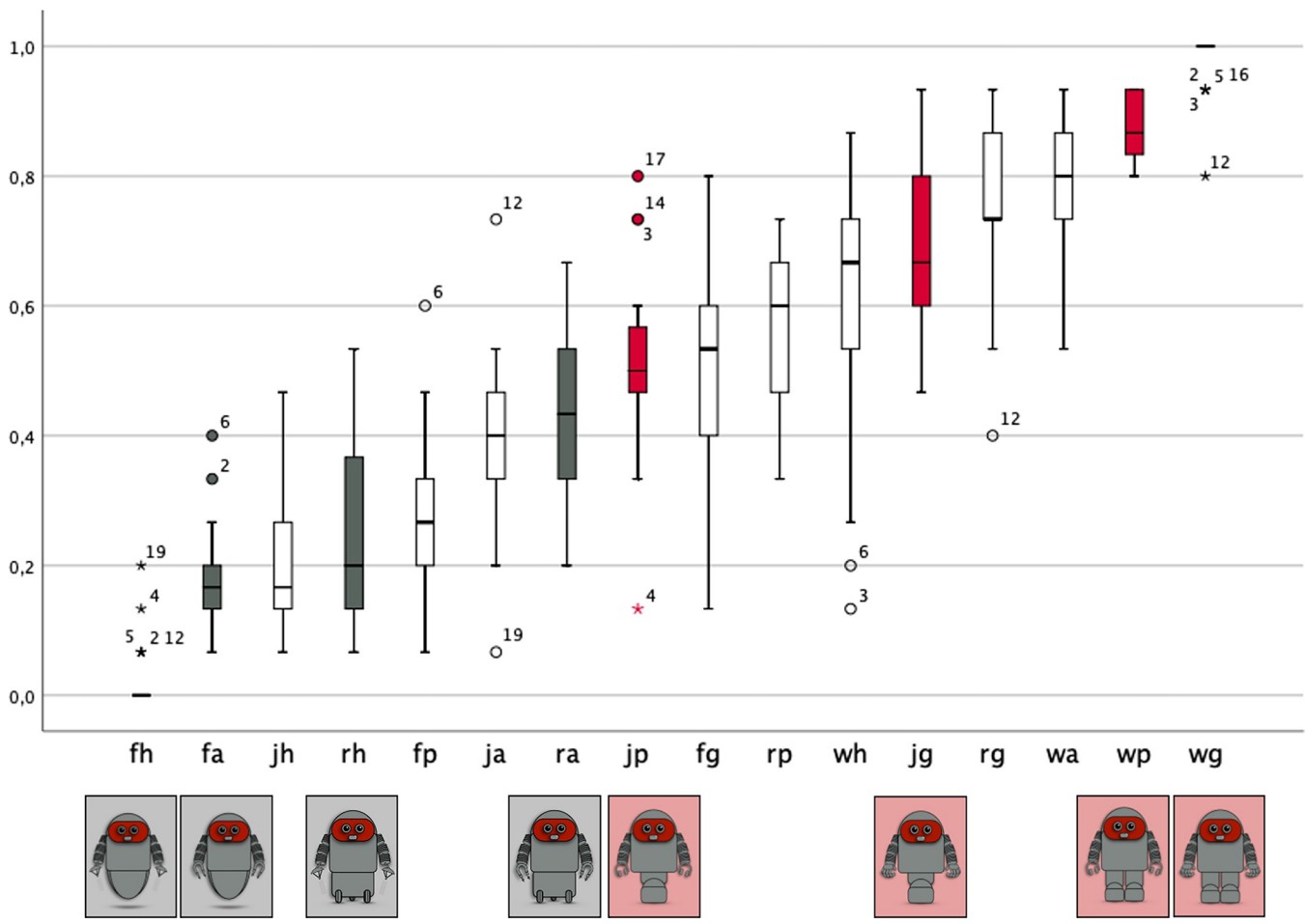

**Fig 4. Estimation of the degree of human resemblance.** Boxplot of the degree of human resemblance for each of the 16 robots (displayed from the lowest (left) to the highest (right) degree pf resemblance). The 2×4 robots with consistent functions are marked with their corresponding picture. Note: Gray symbols = robots with consistent non-simulatable functions of the upper and lower extremities; red symbols = robots with consistent simulatable functions of the upper and lower extremities; white symbols = hybrid robots that combine simulatable and non-simulatable functions. w = walking; j = jumping; r = rolling; f = floating; g = gripping; p = pinching; a = attracting; h = hovering.

scores (ordinal scale). Gray symbols denote results for the four robots that consistently combine upper/lower extremities with non-SIM functions ((floating-hovering), (floating-attracting), (rolling-hovering), (rolling-attracting)), red symbols for the four robots that consistently combines upper/lower extremities with SIM functions ((jumping-pinching), (jumping-gripping), (walking-pinching), (walking-gripping)). The white symbols denote results for robots with h-SIM functions. Note that robots with non-SIM functions are in the bottom half of the distribution and those with SIM capabilities are in the top half.

The results show that the robot-stimuli with consistent functions differ significantly in their degree of perceived human resemblance, with higher values for robots with consistent SIM functions compared to those with consistent non-SIM functions. For robots with consistent SIM functions the lowest score was given to the agent that combines the (jumping-pinching) functions, i.e. the two functions which, if they should be performed with the human body, imply less than optimal performance. The highest score was given to the robot that combines the two functions most characteristic of the human body, i.e., (walking-gripping). The other two robots with consistent SIM functions, i.e., (jumping-gripping) and (walking-pinching)

score between these two extremes. For robots with consistent non-SIM functions, the lowest score was given to the robot that combines the (floating-hovering) functions, i.e. the two functions humans are less familiar with. The highest score was given to the robot that combines the (rolling-attracting) functions, i.e., functions that are familiar to humans.

In the following analyses simulatability and the degree of human resemblance will serve as independent variable to predict the perception of essentialized human qualities and social acceptance.

## 2. Perception of essentialized human qualities

Fig 5 displays the degree of perceived essentialized human qualities (average of the five qualities) as a function of the robots' degree of human resemblance. The attached table gives the same data in digits. The figure suggests that for robots with consistent SIM function (red symbols) the degree of perceived essentialized human qualities increases as human resemblance increases, while the reversed relationship is evident for robots with consistent non-SIM functions (gray symbols). For the hybrid robots (white symbols) some follow the trend described by robots with SIM functions, others follow the trend described by robots with non-SIM functions. In general, hybrid robots with a human-like functions that signals less efficient performance (denoted by the letters 'j' or 'p' for jumping and pinching) scores lower than robots with a human-like functions that signals more efficient performance (denoted by the letters 'w' or 'g' for walking and gripping). Note, the two hybrid robots that combine one of the most (walking; gripping) and one of the least human-like functions (floating; hovering), i.e. robots "fg" and "wh", score highest among all robots with hybrid functions.

To determine predictors for essentialized human qualities, we built a full model of the dependent variable *qualities*, with *simulatability* and *resemblance* as the independent variables (qualities ~ simulatability + resemblance + simulatability × resemblance), where × denotes the interaction between two independent variables. The sample size of the linear models was 2000: 16 robots × 5 qualities × 25 participants. For the full model, type III $F$ tests were performed using R's car::Anova function [45].

Table 1 gives the results for this analysis for the group of all 16 robots, for the group of 8 robots with consistent SIM and consistent non-SIM functions, and for the groups of 4 robots with consistent SIM or consistent non-SIM functions. For the 16 robots, all variables are significant predictors of *qualities*. For the 8 target robots, *simulatability* is a significant predictor of *qualities* ($F_{(1, 996)}$ = 94.957, $p < .001$) with overall higher performance for robots with consistent SIM functions. The significant interaction between *simulatability* and *resemblance* ($F_{(1, 996)}$ = 170.557, $p < .001$) further show that human resemblance has a different effect on the attribution of essentialized human qualities, depending on whether the robots possess consistent SIM or consistent non-SIM functions. For the 4 robots with SIM functions, the degree of essentialized human qualities decreases as human resemblance decreases ($F_{(1, 498)}$ = 129.960, $p < .001$). For the 4 robots with non-SIM functions the degree of essentialized human qualities increases ($F_{(1, 498)}$ = 59.200, $p < .001$).

In short, the attribution of essentialized human qualities obeys different rules depending on whether the robot has mechanisms/functions that can be simulated with the human body. For robots with consistent SIM functions the attribution of essentialized human qualities drops as human resemblance drops, but for robots with consistent non-SIM functions it increases.

## 3. Social acceptance

Fig 6 plots the results of participants' response choices in the social acceptance experiment as a function of human resemblance. The attached table gives the data in digits. Since 'exclude' and 'attack' responses were highly correlated they were added and analyzed together.

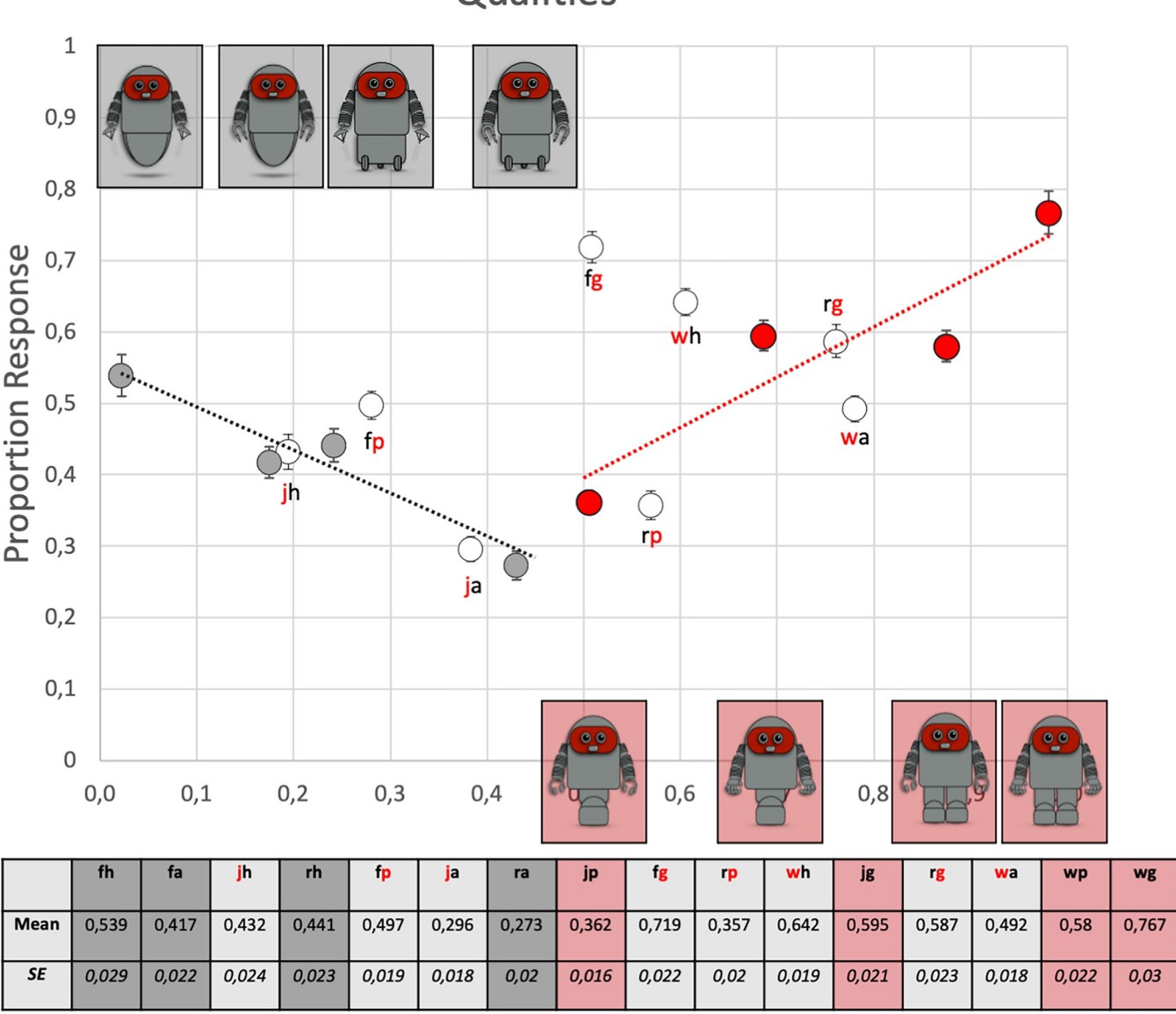

**Fig 5. Estimation the degree of essentialized human qualities (average of the five qualities) for each of the 16 individual robots, presented as a function of the degree of human resemblance.** Error bars are standard errors (SE). The 2x4 robots with consistent functions are marked with their corresponding picture. The color code of the letters in the figure indicate the simulatable function (red) and the non-similatable function (gray) of robots with hybrid functions. The table gives the same data in digits. Note: Gray symbols = robots with consistent non-simulatable functions of the upper and lower extremities; red symbols = robots with consistent simulatable functions of the upper and lower extremities; white symbols = hybrid robots that combine simulatable and non-simulatable functions. w = walking; j = jumping; r = rolling; f = floating; g = gripping; p = pinching; a = attracting; h = hovering.

To test the effect of robot-characteristics on social responses ('inclusion', 'exclusion/abuse' and 'ignore') we used a generalized linear mixed model (GLMM) with a logit link as implemented in R's lme4::glmer (Bates et al., 2015). For the GLMM models, type III Wald chi-squared tests were conducted per effect. Since responses ('include', 'exclude/attack', 'ignore') were exhaustive and we were primarily interested in the acceptance of robots, we carried out the analyzes in two phases: First, we coded 'ignore' vs. 'not ignore' responses as two levels in

**Table 1. Results of the ANOVA for essentialized human qualities.** The left column indicates the robot group over which the regression was conducted. The second column indicates the main effects and interaction, where interaction refers to the interaction of the independent variables of the corresponding full model. Plausible effects were tested on different robot groups (all 16 robots; 8 robots with consistent functions; 4 with either SIM or non-SIM functions).

|  | Effects | df | F |  | ΔR2 |
|---|---|---|---|---|---|
| All robots (16) | Resemblance | 1, 1994 | 7.488 | ** | .003 |
|  | Simulatability | 2, 1994 | 53.365 | *** | .045 |
|  | Interaction | 2, 1994 | 86.859 | *** | .075 |
| Consistent robots (8) | Resemblance | 1, 996 | 0.490 |  | .000 |
|  | Simulatability | 1, 996 | 94.957 | *** | .074 |
|  | Interaction | 1, 996 | 170.557 | *** | .134 |
| SIM robots (4) | Resemblance | 1, 498 | 129.960 | *** | .205 |
| non-SIM robots (4) | Resemblance | 1, 498 | 59.200 | *** | .105 |

Note: df = degrees of freedom; resemblance = human resemblance; quality = essentialized human qualities; SIM: simulatable

(*) $p < .05$

(**) $p < .01$

(***) $p < .001$.

the dependent variable and performed the GLMM analyzes. In the second phase, 'ignore' responses were removed from the data set and 'include' vs. 'non-include' (i.e., 'exclude/attack') responses were coded as two levels in the dependent variable.

For the analyses of the group of 16 robots, a given response (e.g. 'inclusion'), which was represented as a binary variable (chosen or not on each trial), was modeled as a logistic function of fixed effects, i.e. *resemblance*, *simulatability*, *qualities*, and the interaction between *simulatability* and *resemblance*. *Resemblance* and *qualities* are continuous variables, *simulatability* was divided into 2 binary categorical variables: the *simulatability of upper* and the *simulatability of lower extremities* (coded as: simulatable = 1 and non-simulatable = 0). For the analyses of the group of 8 robots and the groups of 4 robots with consistent SIM or non-SIM functions, the simulatability of upper and lower extremities were combined into one single binary variable. Data inspection showed that the variances of random intercepts and random slopes for *resemblance* were generally much larger than for *simulatability* and *qualities*. Therefore, we used random intercepts and random slopes for *resemblance* in by-participant and random intercepts in by-group effects as a basis for our random effects structure. The random slopes for *qualities* in by-participant were included to maximize the random effects structure whenever possible. Random slopes for interactions were not considered because interactions do not substantially vary between participants. Note that all random effects were selected to keep a maximum possible random effects structure. The following variables were thus modeled as random effects. For the group of 16 robots: the *age of the individuals the robot is approaching* in the clip (i.e. the groups of kids, adolescents, young adults, and elderly people) with random intercepts; *participants* with random intercepts, random slopes for *resemblance*, random slopes for *quality*. For the group of 8 robots with consistent upper- and lower-functions: the *age of the individuals the robot is approaching* in the clip (i.e. the groups of kids, adolescents, young adults, and elderly people) with random intercepts; *participants* with random intercepts, random slopes for *resemblance*. However, for 'include' responses random slopes for *resemblance* were dropped from the analyses to enable the model to converge. For the groups of 4 robots with SIM functions: the *age of the individuals the robot is approaching* in the clip (i.e. the groups of kids, adolescents, young adults, and elderly people) with random intercepts; *participants* with random intercepts, random slopes for *resemblance*, random slopes for *quality*. However, for 'include' responses random slopes for *quality* were dropped from the analyses to enable the

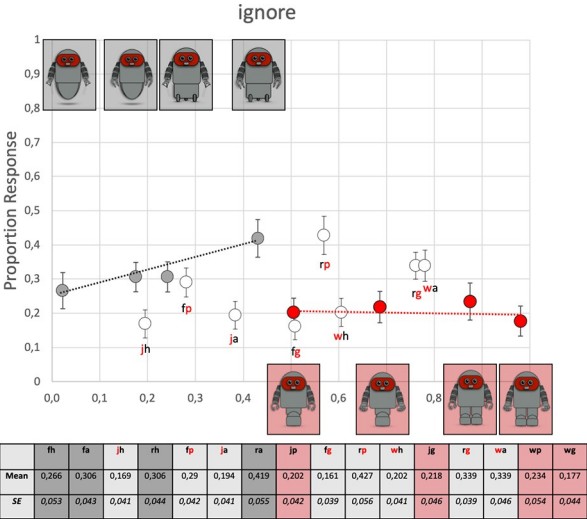

| | fh | fa | jh | rh | fp | ja | ra | jp | fg | rp | wh | jg | rg | wa | wp | wg |
|---|---|---|---|---|---|---|---|---|---|---|---|---|---|---|---|---|
| Mean | 0,266 | 0,306 | 0,169 | 0,306 | 0,29 | 0,194 | 0,419 | 0,202 | 0,161 | 0,427 | 0,202 | 0,218 | 0,339 | 0,339 | 0,234 | 0,177 |
| SE | 0,053 | 0,043 | 0,041 | 0,044 | 0,042 | 0,041 | 0,055 | 0,042 | 0,039 | 0,056 | 0,041 | 0,046 | 0,039 | 0,046 | 0,054 | 0,044 |

Human Resemblance

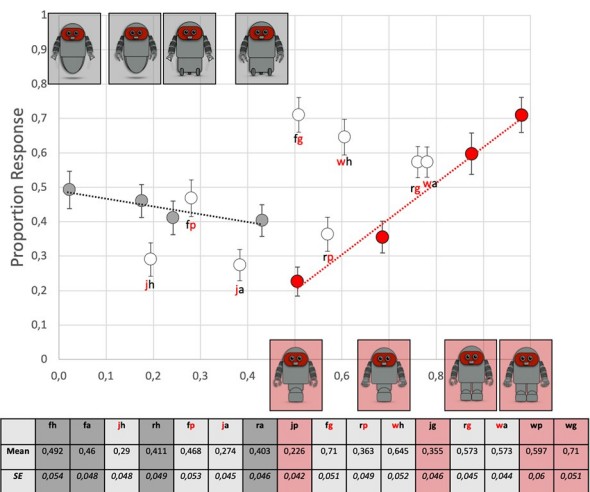

| | fh | fa | jh | rh | fp | ja | ra | jp | fg | rp | wh | jg | rg | wa | wp | wg |
|---|---|---|---|---|---|---|---|---|---|---|---|---|---|---|---|---|
| Mean | 0,492 | 0,46 | 0,29 | 0,411 | 0,468 | 0,274 | 0,403 | 0,226 | 0,71 | 0,363 | 0,645 | 0,355 | 0,573 | 0,573 | 0,597 | 0,71 |
| SE | 0,054 | 0,048 | 0,048 | 0,049 | 0,053 | 0,045 | 0,046 | 0,042 | 0,051 | 0,049 | 0,052 | 0,046 | 0,045 | 0,044 | 0,06 | 0,051 |

Human Resemblance

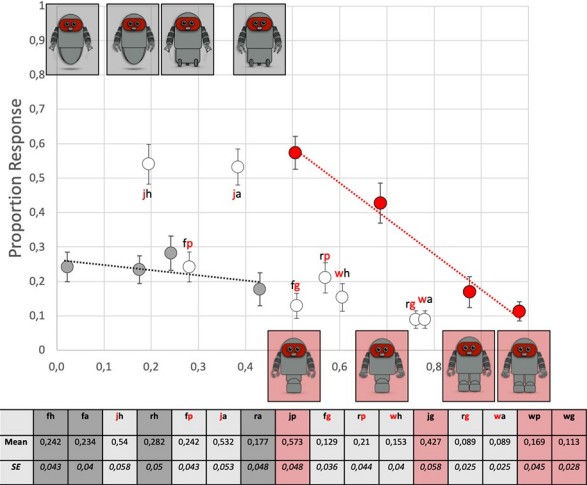

| | fh | fa | jh | rh | fp | ja | ra | jp | fg | rp | wh | jg | rg | wa | wp | wg |
|---|---|---|---|---|---|---|---|---|---|---|---|---|---|---|---|---|
| Mean | 0,242 | 0,234 | 0,54 | 0,282 | 0,242 | 0,532 | 0,177 | 0,573 | 0,129 | 0,21 | 0,153 | 0,427 | 0,089 | 0,089 | 0,169 | 0,113 |
| SE | 0,043 | 0,04 | 0,058 | 0,05 | 0,043 | 0,053 | 0,048 | 0,048 | 0,036 | 0,044 | 0,04 | 0,058 | 0,025 | 0,025 | 0,045 | 0,028 |

Human Resemblance

**Fig 6. (a)** The proportion of 'ignore responses, **(b)** 'include' responses, and **(c)** exclude/attack' responses for each of the 16 individual robots, presented as a function of the degree of human resemblance. Error bars are standard errors (SE). The 2×4 robots with consistent functions are marked with their corresponding picture. The color code of the letters in the figure indicate the simulatable function (red) and the non-simulatable function (gray) of robots with hybrid functions. The table gives the same data in digits. Note: Gray symbols = robots with consistent non-simulatable functions of the upper and lower extremities; red symbols = robots with consistent simulatable functions of the upper and lower extremities; white symbols = hybrid robots that combine simulatable and non-simulatable functions. w = walking; j = jumping; r = rolling; f = floating; g = gripping; p = pinching; a = attracting; h = hovering.

model to converge. Underline{For the groups of 4 robots with non-SIM functions}: the *age of the individuals the robot is approaching* in the clip (i.e. the groups of kids, adolescents, young adults, and elderly people) with random intercepts; *participants* with random intercepts, random slopes for *resemblance*. However, for 'include' responses random slopes for *resemblance* were dropped from the analyses to enable the model to converge. Note, since the correlations between the random intercepts and the random slopes for quality and for resemblance were at ceiling (~1.00), we only used the random intercepts of the by-group effect.

Table 2A summarizes the results for the analyses of the group of 16 robots, Table 2B for the group of 8 robots and the groups of 4 robots with consistent SIM or non-SIM functions.

**Ignore responses.** The upper panel in Fig 6 gives the proportion of 'ignore' responses as a function of human resemblance and shows that on average, robots with consistent non-SIM functions triggered slightly more 'ignore' responses than robots with consistent SIM functions. The GLMM analysis indicates that for the group of 16 robots (Table 2A), *resemblance*, *qualities* and *simulatability of the lower extremities* are significant predictors of 'ignore' responses (note that the absence of an effect of *simulatability of the upper extremities* could stem from the fact that in the clip showing the robot approaching a group of people, the upper limbs were kept motion-less to prevent the robot from appearing agitated). However, in the group of 8 robots and the groups of 4 robots with consistent SIM and non-SIM functions no significant effects were found (Table 2B). Hence, for robots with consistent functions of upper and lower extremities, 'ignore' responses did not vary significantly as a function of the robots' characteristics.

**Include responses.** The left column in Fig 6 plots the proportion of 'include' responses as a function of human resemblance. Note that these data vary with human resemblance in a strikingly similar way as the degree of essentialized human qualities in Fig 5. The acceptability of robots with SIM functions (red symbols) drops as human resemblance drops. For robots with non-SIM functions (black symbols) the reversed pattern is seen, i.e., higher scores for robots that least resemble humans. Like for essentialized human qualities, some of the hybrid robots (white symbols) follow the trend described by robots with SIM functions, others follow the trend described by robots with non-SIM functions. Hybrid robots with a human-like functions that signals less efficient performance (denoted by the letters 'j' or 'p' for jumping and pinching) scores lower than robots with a human-like functions that signals more efficient performance (denoted by the letters 'w' or 'g' for walking and gripping). Finally, the two hybrid robots that combine one of the most (walking; gripping) and one of the least human-like functions (floating; hovering), i.e. robots "fg" and "wh", score highest among all robots with hybrid functions. The GLMM analysis for the 16 robots support this correlation by showing that besides *resemblance*, and *simulatability of the lower extremities*, *qualities* is a significant predictor of 'include' responses (Table 2A). The GLMM analysis for the group of 8 robots with consistent SIM and non-SIM functions further indicates that *resemblance*, *simulatability* and the interaction between *resemblance* × *simulatability* are significant predictors of the 'include' response for robots with consistent functions of the upper and lower extremities. The separate analyses of the group of 4 robots with SIM and non-SIM functions, specify that *resemblance* is

**Table 2. a. GLMM table for the main effects of *resemblance*, *quality* and *simulatability*, and the interaction between (resemblance × simulatability), on social acceptance.** The Effects column denotes main effects on the dependent variable (DV) and interactions (denoted by × marks). The upper part of the table gives results for the 'ignore' versus 'non-ignore' responses based on the total sample; the lower part for 'inclusion' versus 'non-inclusion' responses based on the sample in which 'ignore' responses were discarded. The χ² statistic is calculated from type III Wald's chi-sqaured test between the full model and a model with corresponding variables being restricted. Note: response/n = the ratio between the number of the corresponding DV responses and the total sample; b = coefficient of the corresponding variable; SE = the standard deviations of the corresponding coefficient; resemblance or res. = human resemblance; quality = essentialized human quality; sim. = simulatability; † *p* < .1, * *p* < .05, ** *p* < .01, *** *p* < .001. b. GLMM table for the main effects of *resemblance*, *quality* and *simulatability*, and the interaction between (resemblance × simulatability), on social acceptance. The Effects column denotes main effects on the dependent variable (DV) and interactions (denoted by × marks). The three columns to the right of the Effects column indicate the group of robots included in the test. The upper part of the table gives results for the 'ignore' versus 'non-ignore' responses based on the total sample; the lower part for 'inclusion' versus 'non-inclusion' responses based on the sample in which 'ignore' responses were discarded. The χ² statistic is calculated from type III Wald's chi-sqaured test between the full model and a model with corresponding variables being restricted. Note: response/n = the ratio between the number of the corresponding DV responses and the total sample; b = coefficient of the corresponding variable; SE = the standard deviations of the corresponding coefficient; resemblance or res. = human resemblance; quality = essentialized human quality; sim. = simulatability; SIM = simulatable; † *p* < .1, * *p* < .05, ** *p* < .01, *** *p* < .001.

| DV | Effects | All robots (16) | | | | |
| --- | --- | --- | --- | --- | --- | --- |
| | | Response/n | *b* | *SE* | *χ²* | |
| Ignore | Resemblance | 527/1984 | 1.619 | 0.522 | 9.603 | ** |
| | Quality | | -0.164 | 0.050 | 10.904 | *** |
| | Sim. upper | | 0.058 | 0.223 | 0.067 | |
| | Sim. lower | | -0.931 | 0.222 | 17.615 | *** |
| | Res. × sim. upper | | -0.482 | 0.388 | 1.544 | |
| | Res. × sim. lower | | 0.553 | 0.395 | 1.954 | |
| Inclusion | Resemblance | 936/1457 | 3.838 | 0.571 | 45.135 | *** |
| | Quality | | 0.265 | 0.050 | 28.009 | *** |
| | Sim. upper | | -0.506 | 0.266 | 3.631 | † |
| | Sim. lower | | -1.434 | 0.250 | 32.974 | *** |
| | Res. × sim. upper | | -0.851 | 0.479 | 3.162 | † |
| | Res. × sim. lower | | 0.512 | 0.484 | 1.120 | |

| DV | Effects | Consistent robots (8) | | | | | SIM robots (4) | | | | | nSIM robots (4) | | | |
| --- | --- | --- | --- | --- | --- | --- | --- | --- | --- | --- | --- | --- | --- | --- | --- |
| | | Response/n | b | SE | χ² | | Response/n | b | SE | χ² | | Response/n | b | SE | χ² |
| Ignore | Resemblance | 264/992 | 0.980 | 0.655 | 2.241 | | 103/496 | -2.089 | 3.420 | 0.373 | | 161/496 | 0.632 | 3.260 | 0.038 |
| | Quality | | -0.096 | 0.130 | 0.544 | | | 0.064 | 0.232 | 0.075 | | | -0.185 | 0.319 | 0.336 |
| | Simulatability | | -0.337 | 0.836 | 0.162 | | | | | | | | | | |
| | Res. × sim. | | -0.842 | 2.031 | 0.172 | | | | | | | | | | |
| Inclusion | Resemblance | 453/728 | 4.027 | 0.617 | 42.668 | *** | 234/393 | 10.197 | 2.014 | 25.647 | *** | 219/335 | -3.436 | 3.706 | 0.859 |
| | Quality | | -0.243 | 0.140 | 3.010 | † | | -0.220 | 0.161 | 1.852 | | | -0.438 | 0.374 | 1.372 |
| | Simulatability | | -5.274 | 0.895 | 34.743 | *** | | | | | | | | | |
| | Res. × sim. | | 8.264 | 2.185 | 14.306 | *** | | | | | | | | | |

a predictor of 'include' responses for robots with SIM functions but not for robots with non-SIM functions (Table 2B).

## Discussion

All together, our results can be summarized as follow. In line with our hypothesis, for ASAs with consistent functions of the lower and upper extremities, human resemblance scores were generally higher when the functions could be simulated with the human body (robots with SIM functions > robots with non-SIM functions). Moreover, for such simulatable functions, lower scores were assigned to those that signal less efficient performance if it were to be performed with the human body (jumping/pinching < walking/gripping). Such gradation was also seen for robots with non-SIM functions, with lower human resemblance scores given to functions that were less familiar to humans (floating/hovering < rolling/attracting). By

contrast, contrary to our expectation, the attribution of essentialized human qualities was not correlated with the degree of human resemblance but obeyed different rules depending on whether the robot possessed mechanisms/functions that could be simulated with the human body: For ASAs with consistent SIM functions the attribution of essentialized human qualities increased as human resemblance scores increased. By contrast, for ASAs with consistent non-SIM functions the attribution of essentialized human qualities decreased as human resemblance scores increased. Critically, ASA's acceptability was directly predicted by the degree of perceived essentialized human qualities. These findings are thus consistent with the idea that the degree of simulatability of ASAs' mechanism/functions with the human body plays a role in human attitudes towards social robots. Mechanism/functions that can be simulated might invite the establishment of dominance hierarchies, and ASAs with such mechanism/functions could be downranked if they signal inferior performance compared to humans. As a consequence, these ASAs are more likely to trigger dominance accompanying behavior (intimidation and aggression).

Work on human-robot interaction generally suggests a positive effect of "anthropomorphic designs" on the acceptance of social robots, with anthropomorphism referring primarily to the physical resemblance of robots to humans (see [37]). The higher a robot's resemblance to humans, the higher its acceptance (with the exception of the "uncanny valley" dip that is *not* present in our data, which can be observed for very high but still inferior human resemblance [46]). Our data show, however, that the formula is more complex because the reverse inference is not valid: less resemblance to human does not automatically lead to less acceptance. This is because the notion of human resemblance only makes sense when a reference can be establish to something human (e.g. a one-legged robot looks less like a human because humans have two legs; a robot with a nose-less face looks less like a human because the human face has a nose, etc.). By contrast, non-resemblance is multi-dimensional because an entity can differ from humans in many ways (e.g. what resembles less a human, a tulip, or a cup of coffee?). The claim that increasing "anthropomorphic designs" (physical appearance) are advantageous for ASAs acceptability is true for ASAs with mechanisms/functions that can be referenced to (simulated with) the human body. For ASAs that lack such features, our data instead suggests that "foreignness" ("*floating*" > "*rolling*") might be more beneficial as it increases acceptance and thus helps prevent abuse. Note that our robots with non-SIM mechanism/functions still possess many humanlike features (e.g. a face, a torso, arms) that could serve comparison. Neutralizing these characteristics could thus further increase their social acceptance. Until the physical appearance of ASAs can be engineered to perfectly mimic humans (i.e. when the resemblance to humans is no longer inferior), it therefore seems safer to provide them with mechanisms/functions that cannot be simulated with the human body.

It is important to stress here, that the term "anthropomorphism" does not only refer to the physical appearance but comprise the attribution of character, emotions, intentions etc. to non-human entities [16]. The attribution of essentialized human qualities to ASAs is anthropomorphism. Our results show that unlike physical appearance (resemblance to humans), which differentially affects the acceptability of ASAs depending on the simulatability of mechanism/functions, the attribution of essentialized human qualities directly predicts ASAs acceptability. When the scores for essentialized human qualities are high, the likelihood that the ASAs will be included into a social group is also high. When these scores are low, acceptability is also low. Note though that the present study does not allow identifying elements that drive the attribution of essentialized human qualities to ASAs. Since human resemblance ratings for the ASAs do not correlate with the attribution of essentialized human qualities, the perception of these qualities in robots could be determined by purely functional aspects of the agent (independently of the underlying physical mechanisms). If this is true, the one-legged ASA might

have a higher chance of being accepted if instead of jumping it would float. In addition to replicating findings on interhuman interactions which show that the attribution or denial of human qualities affect social behaviors towards an agent [40], our result underlines that anthropomorphism, in its broader sense, is central in human-ASA interactions.

## Conclusion

Humans are a social species, "hardwired" to respond socially to certain cues (see [47] on human-computer interactions). These social responses are not necessarily also sociocultural desirable. Therefore, introducing novel and artificial social agents into society requires considerations of how they will be received. Our results, along with the accumulating evidence cited in the introduction, indicate that AI, designed for social interactions, can generate sociocultural undesirable behaviors in humans. Controlling for physical aspects in the design of ASAs (e.g. avoiding cues that may provoke avoidance or aggression) can help reduce this risk. However, the triggers of human social behavior go beyond physical aspects. In the 1970s, performer Marina Abramović's art project "Rhythm 0" [48] revealed a nuance in human social behavior that deserves mention in this context. Abramović declared herself to visitors to a museum as an *object* with which they could do whatever they wanted (using items provided by the artist), and this under her sole responsibility. In other words, the actions of the visitors were *exempt from any sanctions*. The artist soon found herself naked, had her skin cut off to drink her blood and ended with a loaded gun pointed at her head. ASAs are purchasable objects whose handling by their owners is (currently) free of sanctions. In human societies, social norms (backed by sanctions) evolve to contain and regulate human conduct for the wellbeing of all e.g. [49,50]. It might be worth considering whether such norms will also be needed for human-ASA societies.

## Author Contributions

**Conceptualization:** Tatjana A. Nazir.

**Formal analysis:** Tatjana A. Nazir, Bing Li.

**Funding acquisition:** Tatjana A. Nazir.

**Investigation:** Benjamin Lebrun.

**Software:** Benjamin Lebrun.

**Writing – original draft:** Tatjana A. Nazir.

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
