## [Decision Letter · Decision Letter 0]

25 Aug 2022

PONE-D-22-09779IMPROVING THE ACCEPTABILITY OF SOCIAL ROBOTS: MAKE THEM LOOK DIFFERENT FROM HUMANSPLOS ONE

Dear Dr. Nazir,

Thank you for submitting your manuscript to PLOS ONE. After careful consideration, we feel that it has merit but does not fully meet PLOS ONE’s publication criteria as it currently stands. Therefore, we invite you to submit a revised version of the manuscript that addresses the points raised during the review process.

We look forward to receiving your revised manuscript.

Kind regards,

Anindita Bhadra, PhD

Academic Editor

PLOS ONE

Journal Requirements:

3. We note that you have stated that you will provide repository information for your data at acceptance. Should your manuscript be accepted for publication, we will hold it until you provide the relevant accession numbers or DOIs necessary to access your data. If you wish to make changes to your Data Availability statement, please describe these changes in your cover letter and we will update your Data Availability statement to reflect the information you provide

Additional Editor Comments (if provided):

This is an interesting topic and I would like to thank you for submitting your work to PLoS ONE. As you would see from the reviewers' comments, one reviewer has raised major concerns about the statistics reported in the manuscript. I agree that you need to revisit the statistics and would recommend a modeling approach. I hope you will be able to revise the manuscript and submit a revised version for consideration.

Reviewers' comments:

Reviewer's Responses to Questions

**Comments to the Author**

1. Is the manuscript technically sound, and do the data support the conclusions?

Reviewer #1: Partly

Reviewer #2: Yes

2. Has the statistical analysis been performed appropriately and rigorously? 

Reviewer #1: No

Reviewer #2: Yes

3. Have the authors made all data underlying the findings in their manuscript fully available?

Reviewer #1: No

Reviewer #2: Yes

4. Is the manuscript presented in an intelligible fashion and written in standard English?

Reviewer #1: Yes

Reviewer #2: Yes

5. Review Comments to the Author

Reviewer #1: This is an interesting paper and certainly timely. My main problems/confusion are with the presented analysis of the data.

Page 11/17: Human resemblance. How exactly is the Human resemblance a 2x2 comparison? In fact, the authors perform two separate one-factorial ANOVAs. I have a couple of problems with understanding this analysis.

I guess the basic problem is that the authors label and treat factors based on the manipulations they did. that is “extremity” (upper, lower) x “type of manipulation” (4 types). This is not really good, because in fact these are only operationalization and not hypotheses-based factors. It also not correct, because it is not really crossed. What the author really want to do is to test hypotheses about simulatablity, efficiency and maybe consistency.

So actually it would be simulatability (high vs. low) vs efficiency (high vs low), although it seems that the authors have no hypotheses about efficiency in low simulatability. However, they offered two different types of low simulatable lower and upper extremities. I hope also with efficiency ideas in mind?! So it would be:

Efficiency

Simulatability Low High

Low Floating / Attracting Rolling / Hovering

High Jumping / Pinching Walking / Gripping

In each cell is the further “technical factor” extremities (upper, lower), so it should be a 2x2x2 ANOVA. The hypotheses for resemblance: Main effect simulatability, maybe main effect efficiency, no main effect for extremity. An interaction effect simul x eff would be possible, if eff is only relevant for high simul, for example (as somewhat hinted at by the authors). To my mind, this is the analysis to be done, not two different one factorial ANOVAS. Another question is, whether the ANOVA is a repeated one? If I understood correctly, each participant produces ratings for each robot through comparison (hence for each factor level). Isn’t that dependent data then? You kind of switch to the robot as unit of analysis (a materials analysis) instead of participants ratings (subject analysis) (maybe because the authors are roboticists rather than social researcher, yet this is social research). In other words, the authors N is not the number of participants, but the number of robots. The variance from different ratings among the participants is lost by this, ins’t it. It is all about the differences among the robots.

A slight problem remaining with this is that it ignores the potential accentuating effects of combining upper and lower extremity. There are consistent robots (both extremities simulatable or not simulatable) and inconsistent robots (mixed extremities). It is not unlikely that this will have an effect and I would include according hypotheses explicitly. In the later analysis of all robot ratings they include this. However, the analysis is hard to follow. I am not sure, where the crucial comparison between consistent and inconsistent robot actually is; all seems reconstructed through post-hoc comparisons. This is also due to not actually introducing the factor “consistency” explicitly. I believe it should be treated as an factor all along. All in all, it makes it very complicated and I believe would have been better maybe not to combine upper and lower. So one possibility would also to only focus on one extremity and to hold the other constant for the analysis above. So for example only use robots that hover and put in the data for all variations of upper extremities in a 2x2 scheme above. However done, it must get more transparent and easier to understand.

Page 12/18. Social acceptability. This analysis should follow the factorial logic outlined above, or at least be always similar to the analysis strategy used in resemblance. In fact, the only difference is the measure (choice versus rating). Again the authors present an analysis with robots as unit of analysis. I find that strange. At the end of the day, each participant chose a certain social response to each level of factor, the robots are only operationalization. So I want to actually know the differences in relative and/or absolute frequencies for each response per factor level and not per robot (even if certain robots operationalize certain factor levels). In a way, the authors change their dependent variable (their measure, the different responses) to their independent variable and the robot ratings to their dependent variable. This is against the experimental logic. In fact, as showed above: the robots are operationalization of factor levels. The ratings of human resemblance are a measure, which should respond to the differences in factor levels (represented by robots). In the same logic, social acceptability should vary with factor levels. Instead the data is analyzed in a correlational manner now, although it originally had been an experiment.

The same happens in the final analysis.

To sum, the purpose of the paper is clear and the initial experimental design is in principle sound. However, the analysis does not follow the logic of the initial design. It mixes factorial and correlational analysis in way that makes it hard to follow. In addition, it remains unclear of whether all results remain stable, if the analysis stays in one logic, since this is very hard to tell from the way the data is presented. This is also due to the fact that descriptives are not reported in a way to understand them in detail (and to use them further in a meta-analysis, for example). The authors chose to present all data in figures, which is of course helpful, but make it impossible to get the true values (one has to guess at least a little). There are “error bars” in the figures, but again …

I believe that the analysis must be completely redone following the same factorial design (see above) for EACH measure and the matching ANOVAS (or frequency statistics, if proportions are problematic) accompanied by an interaction graph.

Reviewer #2: The paper considers the potential links between a social robot’s physical appearance and its acceptability by humans. The authors hypothesize that resemblance of the robots to humans in terms of mechanisms/functions is a crucial to human social behavior towards robots. They studied the relationship between the artificial social agent (ASA) introduced into society and the human behavior towards them. For that purpose, they conducted experiments using computer animated robots and provided extended statistical analysis. The authors state that the group represents either children, teenagers, adults or elderly individuals. Unfortunately, information about the number of participants, participants’ age, and socio-economic background is not provided in the paper. Maybe, the acceptance of the social robots will vary depending on the education level of the participants, too. There is no information in the paper in this respect. Also, the authors do not provide information as to how the participants have been chosen in order to be included in the experimental groups.

It would be a good idea if the authors elaborate more on the interpretation of the results from the statistical analysis.

As a minor technical remark could be mentioned that the text in some figures is difficult to read.

The study proposed in the paper is interesting and the paper is well organized. Although the experiments are conducted with computer animated robots, this study could be useful for the developers of novel social robots, because it shed light on the relationship between the robot design and the level of the acceptance by the humans.

6. PLOS authors have the option to publish the peer review history of their article (what does this mean?). If published, this will include your full peer review and any attached files.

Reviewer #1: No

Reviewer #2: No

---

## [Author Response · Author response to Decision Letter 0]

17 Nov 2022

Please find our response to the reviewer and editor comments in the "Response to Reviewer" letter.

---

## [Decision Letter · Decision Letter 1]

7 Jun 2023

IMPROVING THE ACCEPTABILITY OF SOCIAL ROBOTS: MAKE THEM LOOK DIFFERENT FROM HUMANS

PONE-D-22-09779R1

Dear Dr. Nazir,

We’re pleased to inform you that your manuscript has been judged scientifically suitable for publication and will be formally accepted for publication once it meets all outstanding technical requirements.

Kind regards,

Anindita Bhadra, PhD

Academic Editor

PLOS ONE

Additional Editor Comments (optional):

I am sorry that the reviewing process has taken a ong time, due to the nonresponsive of one of the original reviewers. I have now read the two versions of the manuscript and I agree with Reviewer 2.

Reviewers' comments:

Reviewer's Responses to Questions

**Comments to the Author**

1. If the authors have adequately addressed your comments raised in a previous round of review and you feel that this manuscript is now acceptable for publication, you may indicate that here to bypass the “Comments to the Author” section, enter your conflict of interest statement in the “Confidential to Editor” section, and submit your "Accept" recommendation.

Reviewer #2: All comments have been addressed

2. Is the manuscript technically sound, and do the data support the conclusions?

Reviewer #2: Yes

3. Has the statistical analysis been performed appropriately and rigorously? 

Reviewer #2: Yes

4. Have the authors made all data underlying the findings in their manuscript fully available?

Reviewer #2: Yes

5. Is the manuscript presented in an intelligible fashion and written in standard English?

Reviewer #2: Yes

6. Review Comments to the Author

Reviewer #2: The authors have revised some parts of the paper. It seems that the authors have addressed the reviewer’s comments in the revised version of the paper.

7. PLOS authors have the option to publish the peer review history of their article (what does this mean?). If published, this will include your full peer review and any attached files.

Reviewer #2: No

---

## [Editor Report · Acceptance letter]

11 Aug 2023

PONE-D-22-09779R1 

Improving the acceptability of social robots:
 Make them look different from humans 

Dear Dr. Nazir:

I'm pleased to inform you that your manuscript has been deemed suitable for publication in PLOS ONE. Congratulations! Your manuscript is now with our production department. 

Kind regards, 

on behalf of

Dr. Anindita Bhadra 

Academic Editor

PLOS ONE